# Cystine/Glutamate antiporter system $x_c^-$ deficiency impairs macrophage glutathione metabolism and cytokine production

**Axel de Baat**[1,2]*, **Daniel T. Meier**[1,2], **Adriano Fontana**[1,2], **Marianne Böni-Schnetzler**[1,2], **Marc Y. Donath**[1,2]

1 Clinic of Endocrinology, Diabetes and Metabolism, University Hospital Basel, Basel, Switzerland,
2 Department of Biomedicine, University of Basel, Basel, Switzerland

* axel.debaat@unibas.ch

**Data Availability Statement:** Data is publicly available on Zenodo under DOI: 10.5281/zenodo.8211368.

## Abstract

System $x_c^-$, encoded by *Slc7a11*, is an antiporter responsible for exporting glutamate while importing cystine, which is essential for protein synthesis and the formation of thiol peptides, such as glutathione. Glutathione acts as a co-factor for enzymes responsible for scavenging reactive oxygen species. Upon exposure to bacterial products, macrophages exhibit a rapid upregulation of system $x_c^-$. This study investigates the impact of Slc7a11 deficiency on the functionality of peritoneal and bone marrow-derived macrophages. Our findings reveal that the absence of Slc7a11 results in significantly reduced glutathione levels, compromised mitochondrial flexibility, and hindered cytokine production in bone marrow-derived macrophages. Conversely, system $x_c^-$ has a lesser impact on peritoneal macrophages *in vivo*. These results indicate that system $x_c^-$ is essential for maintaining glutathione levels, mitochondrial functionality, and cytokine production, with a heightened importance under atmospheric oxygen tension.

## Introduction

Macrophages play a crucial role in innate immunity by orchestrating various effector responses against pathogens and maintaining tissue homeostasis. To execute these diverse functions, macrophages must adapt their cellular metabolism in a context-dependent manner. Emerging evidence highlights the importance of metabolic reprogramming in driving macrophage activation, polarization, and effector functions [1]. In particular the mitochondria have been identified as central hubs for metabolic regulation in immune cells, including macrophages [2].

Mitochondria are dynamic organelles that not only generate ATP through oxidative phosphorylation but also regulate reactive oxygen species (ROS) production and apoptosis. Mitochondria additionally play a vital role in the immune response, by functioning as a scaffold for inflammasome assembly [3], augmenting TLR signalling [4], and production of immunomodulatory metabolites [5]. A key component of mitochondrial redox homeostasis is the glutathione system, which consists of reduced glutathione (GSH) and enzymes involved in its synthesis and metabolism [6]. GSH, a tripeptide synthesized from cysteine, glutamate, and

**Funding:** This work was financially supported by Swiss National Science Foundation grant # 310030_18471 received by MYD, https://snf.ch/, The funders had no role in study design, data collection and analysis, decision to publish, or preparation of the manuscript.

**Competing interests:** The authors have declared that no competing interests exist.

glycine, is the most abundant co-factor for redox scavenging proteins and is therefore involved in maintaining redox balance, detoxification, and cellular signaling.

The glutamate-cystine antiporter, known as system x$_c^-$ (Sxc), plays a pivotal role in the glutathione system by mediating the uptake of cystine, which, upon reduction to cysteine, serves as the rate-limiting substrate for GSH synthesis [7]. The Sxc transporter comprises two subunits: the catalytic light chain (SLC7A11) and the regulatory subunit 4F2hc (SLC3A2) [8]. By exchanging extracellular cystine for intracellular glutamate, Sxc enables cystine import, which is then employed for GSH production [9]. A recent study has demonstrated that Sxc deficiency and inhibition with a small molecule inhibitor led to improved wound healing in diabetic mice [10]. It is surprising that interference with anabolic processes in important effectors such as dendritic cells and macrophages improves wound healing. To understand the physiological context, this study aims to elucidate the functional aspects of Sxc in the regulation of macrophage effector responses. To understand the physiological role of Sxc, we used bone marrow-derived macrophages and peritoneal macrophages to study the implications of Sxc deficiency on glutathione homeostasis, mitochondrial metabolism and the inflammatory response to bacterial antigen. Sxc deficiency causes cystine shortage, which results in impaired mitochondrial flexibility and cytokine production.

## Materials and methods

### Mouse models

All animal experiments were conducted according to the Swiss Veterinary Law and Institutional Guidelines and were approved by the Veterinary Office of Basel-Stadt (https://www.veterinaeramt.bs.ch/tierschutz/tierversuche.html). All researchers received training from the Cantonal authorities for the care and handling of the animals. All mice were housed in a temperature-controlled facility with a 12 h light–12 h dark cycle, had free access to food and water (plant-based chow, 3436, Kliba Nafag, Kaiseraugst, Switzerland), and were given plastic housing.

C57Bl6NCrl mice were used as wild-types and were derived from our own in-house colonies. Constitutive *Slc7a11* knockout mice were first generated by Sato et al. on a C57Bl6J background and were maintained as an in-house colony. These mice were generated by inserting a GFP cassette into exon 1 of the *Slc7a11* gene [9]. Animals were propagated by crossing heterozygous parents in order to ensure littermates represented in both experimental groups.

### Endotoxemia

To simulate endotoxemia, mice were injected with 2mg/kg body weight lipoplysaccharide (LPS) (Escherichia coli O26:B6, Sigma-Aldrich, cat. L8274) dissolved in PBS. Mice were monitored every 30 minutes during the 2 hours before sacrifice. The experimental procedure was approved by Veterinary Office of Basel-Stadt (https://www.veterinaeramt.bs.ch/tierschutz/tierversuche.html). Since the LPS dose was low and the animals were euthanized with CO$_2$ 2 hours after injection, no humane endpoints were reached.

### RNA Isolation and quantitative real-time PCR

RNA was isolated with the Nucleo Spin RNA II Kit (cat. 740955, Macherey and Nagel, Germany). cDNA was prepared with the GoScript Reverse Transcription Mix containing random primers (A2801, Promega, Switzerland) according to the manufacturer's instructions. For quantitative real-time PCR (qPCR) of *Chac1*, *Il1b* and *Slc7a11*, SYBRgreen-based chemistry with GoTaq Polymerase (cat. A6002 Promega, Switzerland) and the ABI 7500 fast System was employed (Applied Biosystems, USA). *Hprt1* was used as a housekeeping gene and expression

**Table 1. Primer sequences.**

| Target | Forward 5'->3' | Reverse 5'->3' |
|---|---|---|
| Slc7a11 | GGCACCGTCATCGGATCAG | CTCCACAGGCAGACCAGAAAA |
| Chac1 | CTGTGGATTTTCGGGTACGG | CCCCTATGGAAGGTGTCTCC |
| Chop (Ddit3) | AAGCCTGGTATGAGGATCTGC | TTCCTGGGGATGAGATATAGGTG |
| Xbp1-s | TGAGTCCGCAGCAGGTG | AGATGTTCTGGGGAGGTGAC |
| Il1b | GAAATGCCACCTTTTGACAGTG | TGGATGCTCTCATCAGGACAG |
| Hprt1 | TCAGTCAACGGGGGACATAAA | GGGGCTGTACTGCTTAACCAG |

levels were calculated as (gene of interest – housekeeping gene) so as to not assume an amplification efficiency of 2 and to display the data in the most unadulterated manner. Primer sequences can be found in Table 1 and were chosen from the first ranked pairs in the Harvard Primer Bank (https://pga.mgh.harvard.edu/primerbank/, verified 24-07-23).

## Bone marrow derived macrophages (BMDM)

To prepare bone marrow derived macrophages, mice were euthanized with $CO_2$, after which both hindlegs of 16-26-week-old mice were harvested and stripped to the bone. Then both the tibia and femur were cut on both ends and flushed with DMEM (Cat nr. 11966025, GIBCO, Germany) supplemented with 11.1 mM glucose, 1mM pyruvate, Non- Essential Amino Acids (Cat nr. 11140050, GIBCO, Germany), 100 units/ml penicillin, 100 μg/ml streptomycin, 2 mM glutamine, 50 μg/ml gentamycin, 10 μg/ml Fungison, 50 uM beta-mercapto-ethanol, 50 ng/ml M-CSF (Cat nr. 576408, Biolegend) and 10% FCS (BMDM medium). The cells were then cultured in BMDM medium in 9 cm bacterial culture dishes (Sarstedt, Germany) at a density of $10^7$ cells per dish. Medium changes were performed 3 and 6 days after the start of the culture. At day 8 the cells are harvested by incubation at 4°C for 10 minutes in PBS + 0.5% BSA + 5mM EDTA, washed with fresh medium and subsequently polarized with LPS SM (100ng/mL) (Cat nr. tlrl-smlps, InvitroGen, Germany), IL-4 (20ng/mL) (Cat nr. 715004, Biolegend, UK) or oxidized 1-Palmitoyl-2-arachidonoyl-sn-glycero-3-phosphocholine (oxPAPC) (50ug/mL) (Cat nr. 870604, Avanti, USA) in BMDM medium for 6–24 hours.

For cas9-based disruption of *Chac1*, BMDMs cultured for 4 days were nucleofected with the P3 Primary Cell 4D-Nucleofector (Cat nr. V4XP-3032, Lonza, Switzerland) according to manufacturer's instructions using 4D-Nucleofector. Briefly, $1 \times 10^7$ BMDMs were resuspended in 100 μl of Nucleofector Solution and combined with ribonucleoprotein solution. Mm.Cas9.CHAC1.1.AA was selected from the predesigned CRISPR-Cas9 guideRNAs Tool from Integrated DNA Technologies (IDT, USA). Per reaction, 900 pmol crRNA (IDT) or 900 pmol negative control crRNA #1 were mixed with 900 pmol trRNA in nuclease-free duplex buffer, annealed at 95°C for 5 min and added to 300 pmol Cas9(QB3 MacroLab, UC Berkeley) followed by incubation at room temperature for 10 min. Program CM-137 was used. Cells were then plated and used for subsequent analysis.

## Flow cytometry and cell analysis

Flow cytometry experiments were performed on a CytoFLEX V2-B4-R2 Flow Cytometer (8 Detectors, 3 Lasers) (Beckman Coultier, Germany). In short, cells were harvested as described above and subsequently stained with the appropriate antibodies and dyes. For flow cytometry experiments, BMDMs or peritoneal lavage was harvested. F4/80 APC-Cy7 (Cat nr. 123113, Biolegend, UK) and CD11b PE-Cy7 (Cat nr. 101215, Biolegend, UK) positive cells were considered macrophages.

For measuring intracellular thiol levels, Thioltracker dye (T10095, Thermo Fisher Switzerland), was used according to manufacturer's directions.

**Mitochondrial membrane potential determination.**    To determine membrane potential of the mitochondria, Mitotracker Red CMXRos (Cat nr. M7512, ThermoFisher, Germany) and Mitotracker Green FM (Cat nr. M7514, ThermoFisher, Germany) staining was performed in 100 nM concentration for both dyes according to manufacturer's protocol. Cells were stained in DMEM with 0.5% (w/v) BSA instead of FCS. The cells were not washed after staining and immediately analyzed after incubation in the staining buffer. The average membrane potential per mitochondrion was calculated as mitotracker red/mitotracker green.

**Mitochondrial superoxide production measurement.**    To quantify mitochondrial superoxide production, MitoSox Red (Cat nr. M36008, ThermoFisher, Germany) and Mitotracker Green FM (Cat nr. M7514, ThermoFisher, Germany) staining was performed in 100 nM concentration for both dyes according to manufacturer's protocol. Cells were stained in DMEM (Cat nr. 11966025, GIBCO, Germany), supplemented with BSA instead of FCS, and 11.1mM glucose. The cells were not washed after staining and immediately acquired after incubation in the staining buffer. The average membrane potential per mitochondrion was calculated as mitosox red/mitotracker green.

**Intracellular cytokine measurement.**   Harvested BMDMs or Peritoneal macrophages were fixed with 2% formalin and subsequently permeabilized with 0.3% Triton X-100 dilution in PBS. After permeabilization, macrophages were stained with anti-IL-1b (Cat. 25-7114-82, Thermo Fisher, Germany), anti-TNF (Cat. 506343, Biolegend, UK), and anti-IL-10 (Cat. 505005, Biolegend, UK). During analysis, cells were gated for macrophages and mean fluorescence intensity was measured as representative of intracellular cytokine levels.

## *In vitro* IL-1b/IL-10 secretion assay

250′000 cells/well were cultured overnight in 250 ul of medium in a 96 well plate. Cells were washed with warm PBS and treated for 23 h with or without LPS at a concentration of 100 ng/mL (Cat nr. tlrl-smlps, LPS from S. minnesota R595, InvivoGen, Germany). To further stimulate IL-1b secretion, ATP (Cat nr. tlrl-atpl, InvivoGen, Germany) was added at a concentration of 2 mM for 60 min prior to collection of culture supernatants. IL-1b concentrations in cell culture supernatants were measured using MSD mouse IL-1b assay kit (K152QPD; Meso Scale Discovery) and normalized to protein content as measured by BCA assay (Cat nr. 23225, ThermoFisher, Germany). IL-10 was assessed using a standard IL-10 ELISA according to manufacturer's protocol (Cat nr. 431411, Biolegend, UK).

## ATP, glutathione and redox potential measurements

ATP measurements were performed using CelltiterGlo3D (cat. G9681, Promega, Switzerland). For GSH measurements the glutathione-glo assay (cat. V6911, Promega, Switzerland) was used. For intracellular measurements, 100 000 cells were plated in a well of a translucent bottom 96 well plate, in which the assay was performed according to the manufacturers protocol. For glutathione measurements of medium, 50 uL cultured medium was combined with 2X reaction mix and analysed according to the manufacturer's protocol. Medium redox potential was measured of media samples using antioxidant assay kit (MAK334, Sigma-Aldrich).

## Data analysis

Results were analyzed with Prism 9.1.0 (GraphPad, USA) or Python and Jupyter notebook (Jupyter Labs). p < 0.05 was considered to be statistically significant. Results were expressed as mean ± SEM. Data were analyzed with Wilcoxon signed-rank test. Statistical details of

experiments are described in the figure legends and in the figures. Data will be made available upon publication on Zenodo under DOI: 10.5281/zenodo.8211368.

## Results

### System Xc- is specifically induced by LPS and regulates GSH levels

We initially investigated the transcriptional regulation of the light chain of Sxc, Slc7a11. BMDMs were stimulated with LPS and IL-4, leading to M1 and M2 polarization, respectively. Additionally, oxPAPC was employed to induce the Mox polarization, a recently described redox regulatory phenotype [11]. *Slc7a11* expression was specifically upregulated by LPS, but not by other treatments (Fig 1A). Moreover, Slc7a11 upregulation occurred rapidly within 1 hour, reaching peak expression levels at 4 hours post-stimulation (Fig 1B).

Sxc has been hypothesized to participate in a circular pathway in which oxidized cystine is imported and subsequently exported as GSH. Consequently, we measured the antioxidant potential of the culture medium. In Sxc-deficient (KO) animals, M1 and Mox polarized macrophages exhibited a reduction in reductive potential compared to wild-type (WT) cells (Fig 1C). We then evaluated medium GSH concentrations, discovering a decrease in media collected from KO BMDMs regardless of stimulus. This reduction was partially alleviated by treatment with γ-L-Glutamyl-L-cysteine (GGC), a GSH precursor with higher uptake than GSH (Fig 1D). Intracellular GSH levels were more drastically decreased compared to cell culture medium levels but were also ameliorated by GGC treatment. However, GGC treatment led to lower GSH levels in WT BMDM (Fig 1E). Similarly, total thiol levels were lower in KO BMDMs compared to control BMDMs (Fig 1F).

### Sxc modulates mitochondrial superoxide and membrane potential

As GSH is involved in mitochondrial redox maintenance, we assessed mitochondrial function using flow cytometry based detection of mitochondrial stainings. Mitochondrial superoxide, when normalized to mitochondrial mass as measured by mitotracker green, was increased in KO BMDMs (Fig 1G). We then assessed mitochondrial membrane potential. When assessing BMDMs based on their respective mitotracker red and mitotracker green fluorescence intensity, one can see a bifurcation in populations (Fig 1H). The frequency of the PE-Hi population was increased in LPS treated KO macrophages (Fig 1I), suggesting a dampened mitochondrial response to LPS. The lower mitochondrial potential population(PE-Lo) had higher membrane polarization compared to WT BMDMs(Fig 1J). Conversely, the membrane potential per mitochondrion was lower in KO BMDMs in unstimulated conditions in the high mitotracker (PE-Hi) population (Fig 1K). These results suggest KO BMDMs have higher mitochondrial superoxide and less mitochondrial flexibility in membrane potential in response to LPS stimulus.

### Sxc is required for cytokine production in response to LPS

Cysteine is a proteinogenic amino acid, Sxc deficiency might disrupt cytokine production. To assess the ability of Sxc-deficient BMDMs to produce cytokines, we examined IL-1b and IL-10 secretion, representing canonical pro-inflammatory and anti-inflammatory responses, respectively. IL-1b secretion was reduced in KO BMDMs in response to LPS, and it increased when treated with GGC (Fig 2A). IL-10 secretion was also decreased without any effect from GGC treatment (Fig 2B). We then measured intracellular cytokine levels using antibody staining. Decreased IL-10 and IL-1b levels were observed, and treatment with the reducing agent beta-

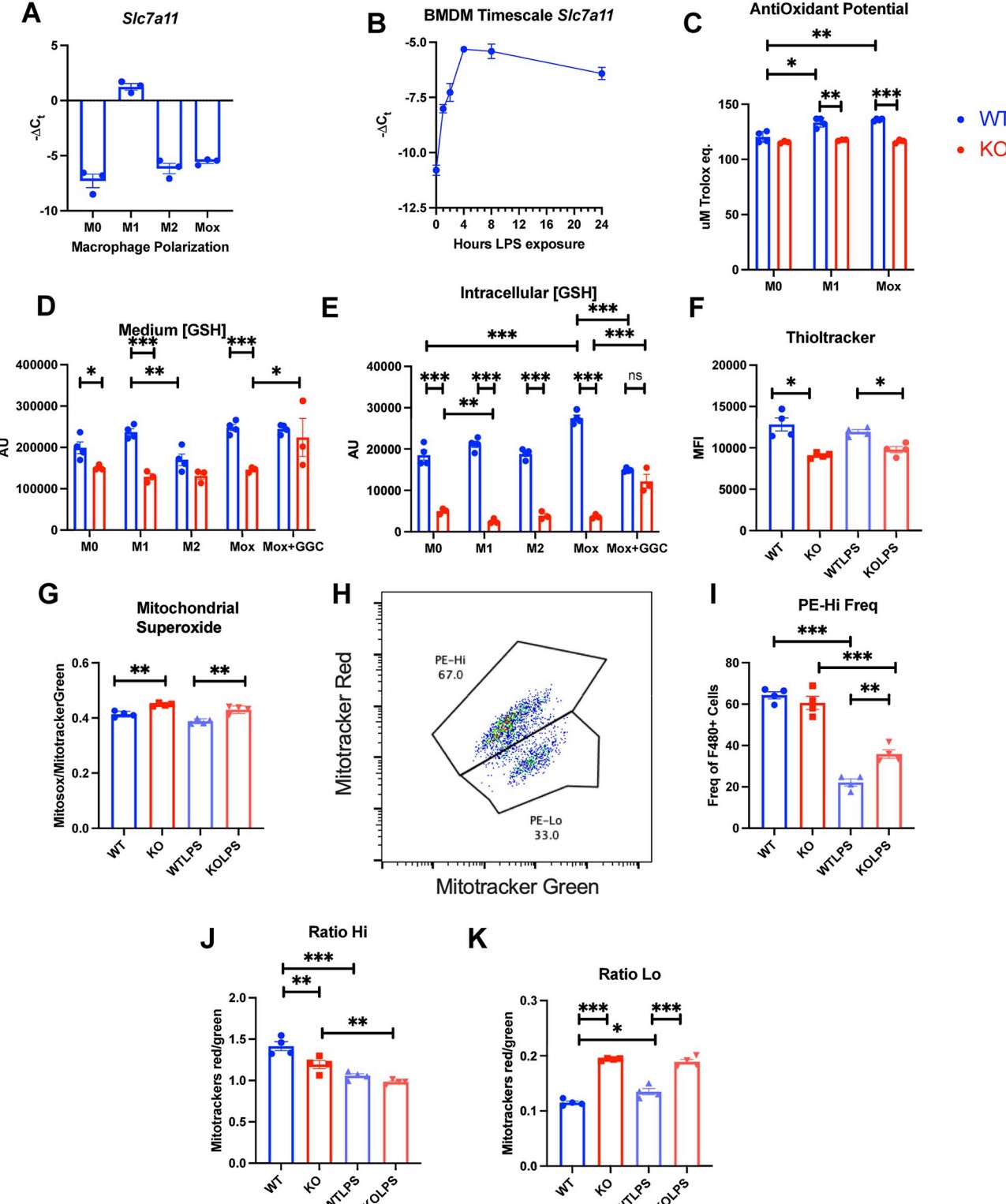

**Fig 1. Slc7a11 is increased in response to LPS, modulates GSH levels and affects mitochondrial membrane potential in BMDMs.** Blue = WT, Red = KO. (A) rtPCR quantification of Slc7a11 expression in different BMDM polarizations. (B) Time dynamics of Slc7a11 in response to LPS stimulus. (C) Medium anti-oxidant potential, (D) Medium GSH levels, (E) Intracellular GSH concentrations. (F) Thiol-dependent fluorescence, (G) Mitochondrial superoxide fluorescence, (H) Representative plot of mitotracker red high and low populations, (I) Frequency of high population compared to low. (J) mitochondrial membrane potential in mitotracker red high population expressed as mitotracker red/ mitotracker green, (K)

mitochondrial membrane potential in mitotracker low population, Each dot represents a BMDM culture from 1 mouse, (B) N = 3. Statistics: two-sided Mann–Whitney U test; error bars represent SD; ★p < 0.05, ★★p<0.01, ★★★p < 0.001.

mercapto-ethanol (BME) increased intracellular cytokines in both KO and WT BMDMs without affecting IL-10 levels (Fig 2C and 2D).

### *Chac1* disruption further exacerbates the decrease in GSH and cytokine production caused by Sxc deficiency

*Chac1* is a gene encoding a glutathione-degrading enzyme that is upregulated in response to declining intracellular cysteine levels. Indeed, we discovered that *Chac1* expression was

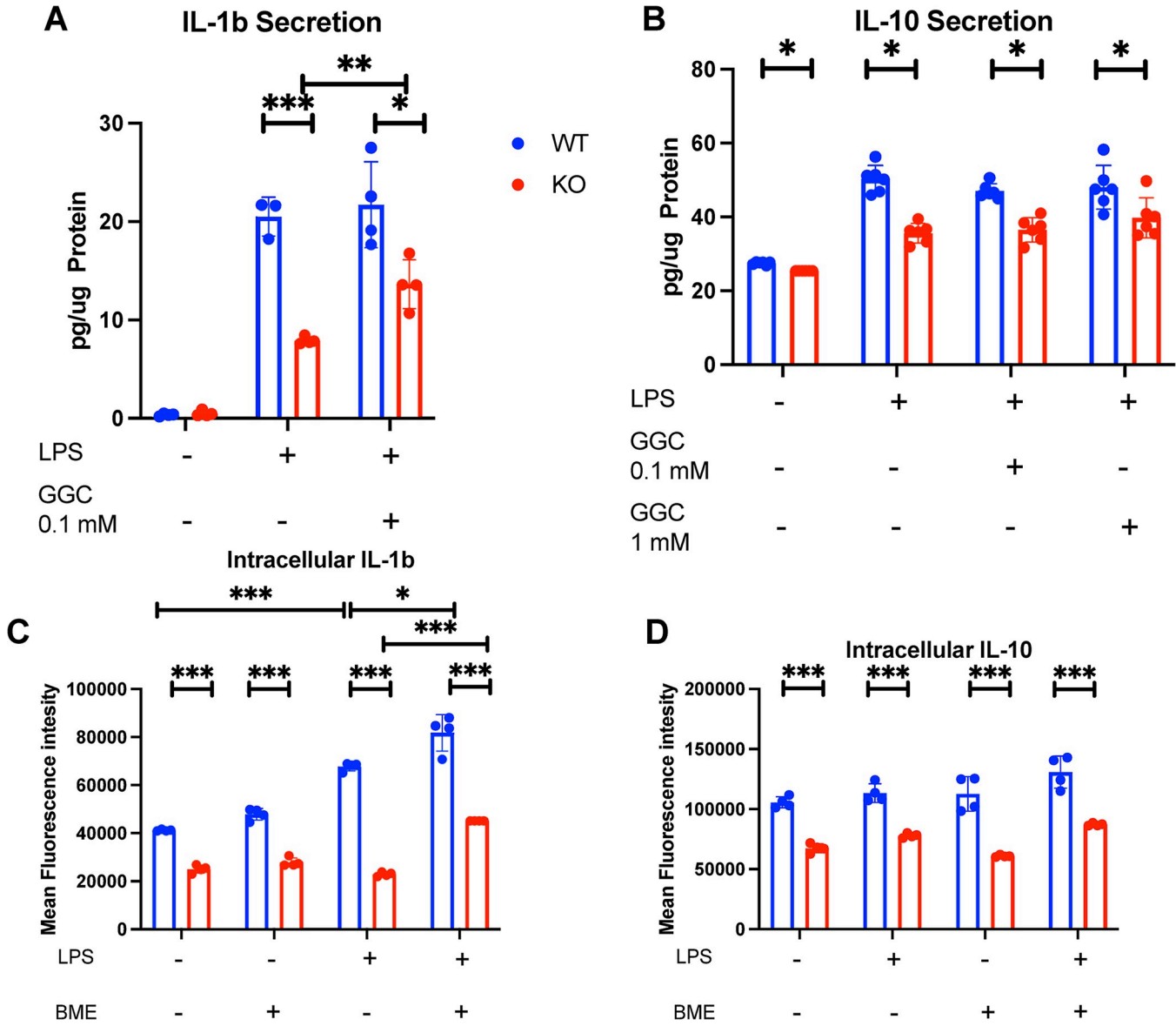

**Fig 2. *In vitro* cytokine secretion and intracellular levels in BMDMs.** (A) IL-1b secretion, (B) IL-10 secretion, (C) intracellular IL-1b levels, BME = beta-mercapto-ethanol, (D) Intracellular IL-10 levels. Each dot represents a BMDM culture from 1 mouse. Statistics: two-sided Mann–Whitney U test; error bars represent SD; ★p < 0.05, ★★p<0.01, ★★★p < 0.001.

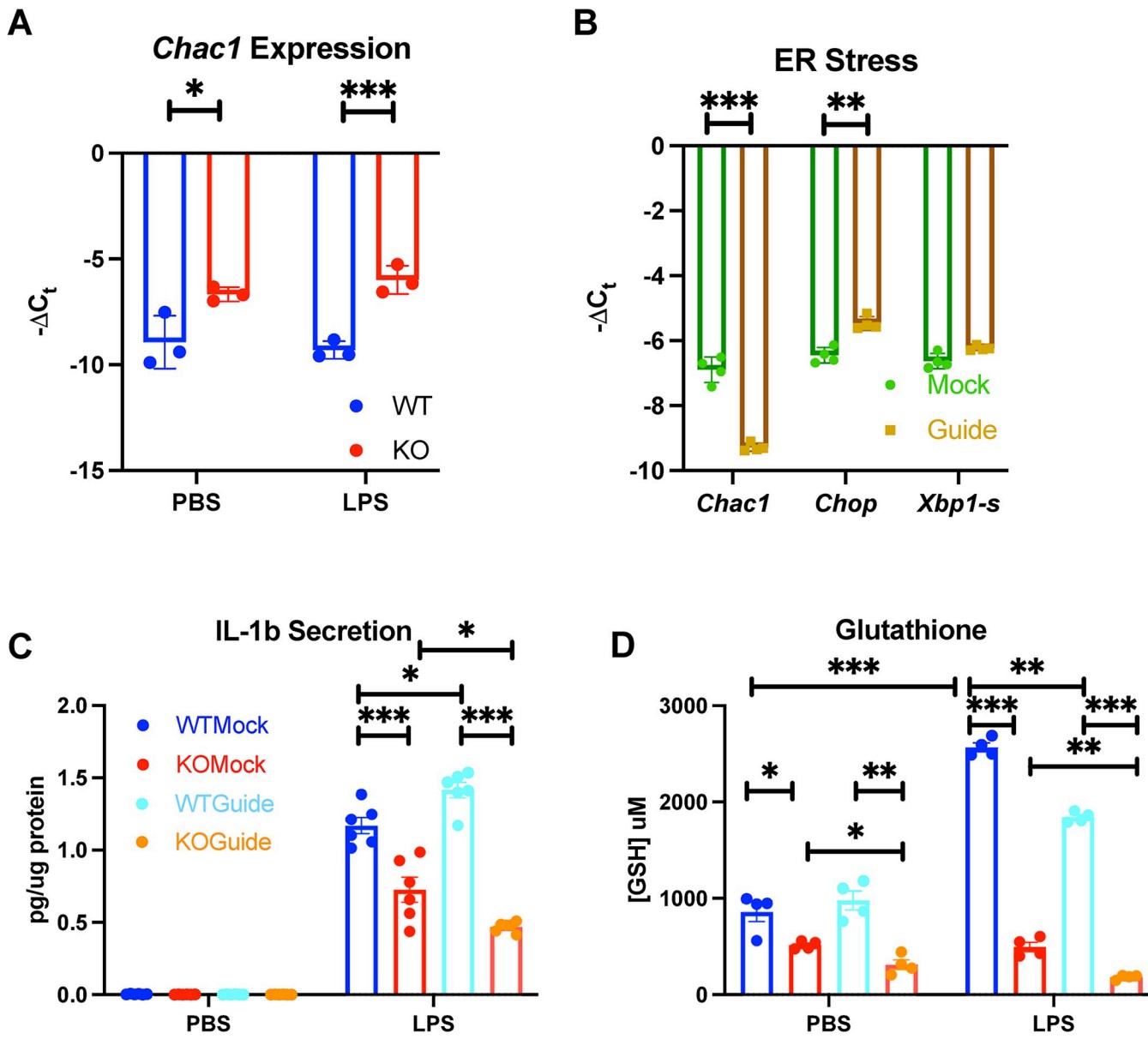

**Fig 3. Implications of Chac1 disruption in BMDMs.** (A) *Chac1* expression in WT and KO BMDMs. (B) Expression of *Chac1* and ER stress markers in response to *Chac1* disruption (C) IL-1b secretion. (D) Glutathione concentrations in WT and KO with and without *Chac1* disruption. Each dot represents a BMDM culture from 1 mouse, guide and mock groups where derived from paired cultures. Statistics: two-sided Mann–Whitney U test; error bars represent SD; ★p < 0.05, ★★p<0.01, ★★★p < 0.001.

chronically elevated in KO BMDMs (Fig 3A). To further investigate *Chac1*'s role and its potential impact on macrophage function, we disrupted the *Chac1* gene using CRISPR-Cas9. Given that the endoplasmic reticulum (ER) relies on strict control of the oxidative environment, we confirmed an approximately 3 cycle threshold (Ct) drop in *Chac1* expression and a slight upregulation of the ER stress marker *Chop* (Fig 3B). We then evaluated WT and KO BMDMs with disrupted *Chac1*. IL-1b production was higher in *Chac1*-disrupted WT BMDMs, while it was lower in KO BMDMs compared to their respective mock transduced counterparts (Fig 3C). GSH concentrations increased with LPS treatment but were reduced by *Chac1* disruption in

both KO and WT BMDMs (Fig 3D). These data suggest that *Chac1* plays a complex role in macrophage function.

## Sxc deficiency in peritoneal macrophages results in a similar phenotype to BMDMs *in vitro*

To validate some of our BMDM findings ex vivo, we isolated cells using peritoneal lavage. After 24 hours of culture and LPS stimulation, IL-1b secretion decreased in KO lavage (Fig 4A). IL-10 secretion was also reduced in both LPS-stimulated and unstimulated conditions (Fig 4B). To examine the temporal dynamics of IL-1b and Chac1 expression in response to LPS stimulus, we performed a 24-hour time course with cells obtained by peritoneal lavage. IL-1b expression rapidly increased in response to LPS and exhibited significant differences

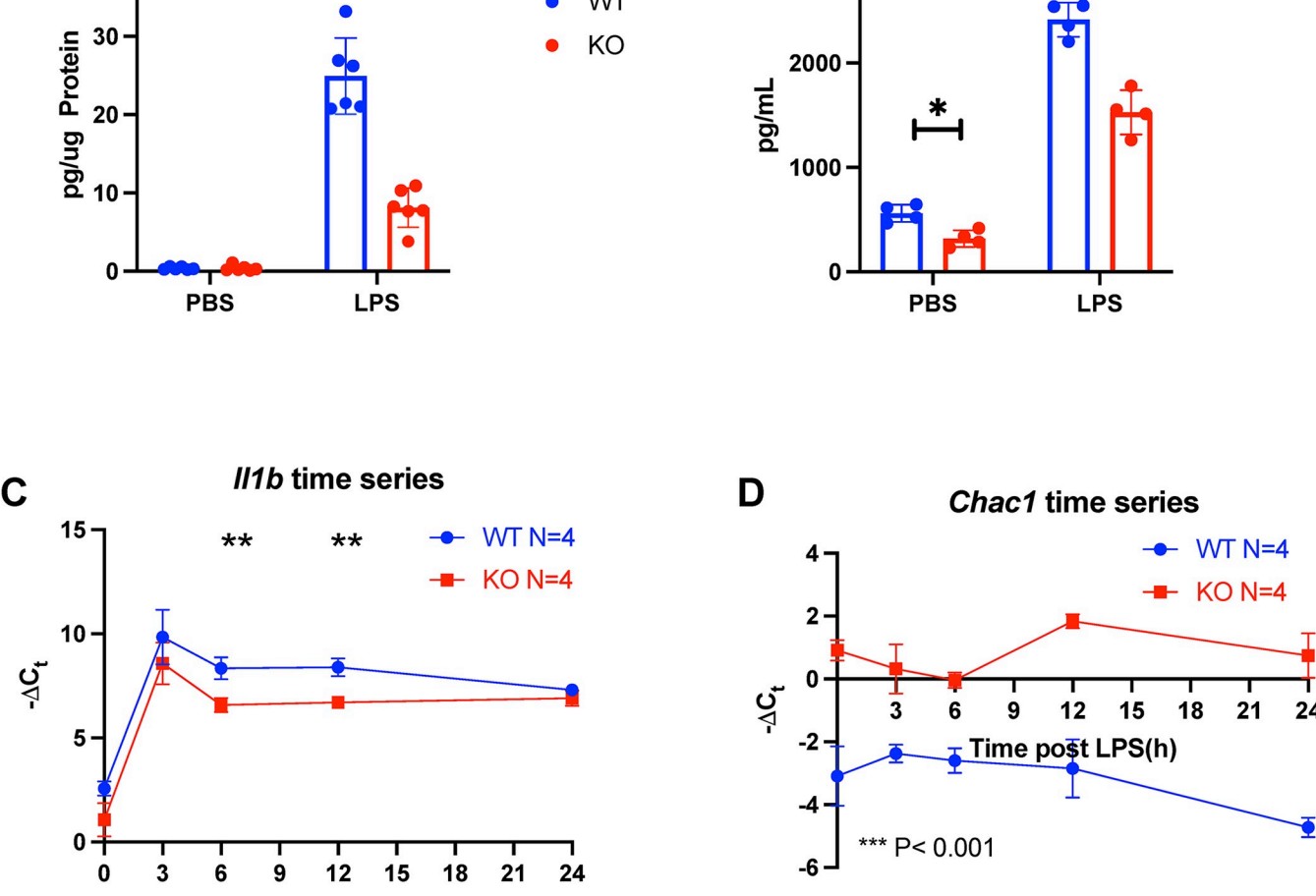

**Fig 4. *In vitro* cytokine production and gene expression in peritoneal macrophages.** (A) IL-1b secretion, (B) IL-10 secretion, (C) *Il1b* and (D) *Chac1* expression of the course of 24 hours. Each dot represents peritoneal lavage from 1 mouse. Statistics: two-sided Mann–Whitney U test; error bars represent SD; ★p < 0.05, ★★p<0.01, ★★★p < 0.001.

between WT and KO genotypes (Fig 4C). *Chac1* expression remained unchanged in response to LPS but was chronically elevated in KO lavage (Fig 4D). These findings align with our observations in BMDMs.

**Peritoneal macrophages isolated after *in vivo* LPS stimulus do not show altered cytokines or mitochondrial activity in response to Sxc deficiency.** We then used Sxc-deficient mice to validate our findings *in vivo* by examining CD11b+, F4/80+ peritoneal macrophages from the peritoneal lavage. In this experimental setup, we minimized the time between isolation and analysis to avoid any effects induced by culturing. Contrary to the *in vitro* results, intracellular IL-1b and IL-10 levels were similar between genotypes (Fig 5A and 5B).

Endotoxemia is a pathological condition characterized by the presence and circulation of endotoxins, typically lipopolysaccharides from the outer membrane of Gram-negative bacteria, in the bloodstream, leading to systemic inflammatory response and potentially severe clinical symptoms. Given the specific *Slc7a11* upregulating properties of LPS, we induced endotoxemia by injection with 2mg/kgbw LPS in WT and KO mice and harvested peritoneal lavage 2 hours after induction to measure intracellular cytokines. IL-1b and IL-10 levels did not differ significantly between KO and control macrophages, despite an increase in response to LPS administration (Fig 5C and 5D); however, the increase was not significant for IL-10 in the KO macrophages (Fig 5D). Total thiol levels were also unchanged between genotypes but increased in response to LPS (Fig 5E). Mitochondrial membrane potential decreased in response to LPS but did not differ between genotypes (Fig 5F). Mitochondrial superoxide production remained unchanged (Fig 5G). In the absence of LPS stimulation, whole peritoneal lavage showed lower ATP and GSH levels in macrophages from KO mice compared to WT mice (Fig 5H and 5I). These data suggest that peritoneal macrophages analyzed rapidly after isolation do not exhibit the same alterations as those observed *in vitro*.

## Discussion

In this investigation, we examined the role of Sxc in macrophage response to LPS, specifically assessing its impact on GSH levels, cytokine production, and mitochondrial function. Our findings indicate that Sxc is rapidly upregulated in response to LPS, with its deficiency leading to altered mitochondrial metabolism and decreased cytokine production. Notably, the effects of Sxc were more prominent *in vitro* due to distinct atmospheric conditions, whereas *in vivo*, these effects were not observed, possibly due to redundancy and compensation in metabolic pathways.

Sxc appears to have a specialized role in the response to LPS in macrophages, as evidenced by its specific and rapid upregulation. Mox macrophages, a redox regulatory macrophage subtype, interestingly do not upregulate Slc7a11 to the same extent as LPS-stimulated macrophages. This suggests that the upregulation of Slc7a11 is driven more by cysteine demand from protein synthesis for the inflammatory response than by redox regulatory purposes. Sxc does contribute to GSH production, as Mox macrophages exhibited increased GSH levels, which did not occur in the absence of Sxc. Macrophages secrete measurable amounts of GSH into the extracellular medium, a function that appears to be diminished in KO macrophages, significantly affecting the total antioxidant potential of the medium.

Given that ROS production is an inevitable consequence of mitochondrial metabolism, lower GSH levels resulting from Sxc deficiency lead to altered mitochondrial metabolism. This is characterized by increased mitochondrial superoxide levels and an apparent reduction in the ability of mitochondria to adjust their membrane potential in response to inflammatory stimuli. Upon LPS activation, macrophages display diminished mitochondrial membrane potential, resulting in two populations separated by membrane potential. In the high

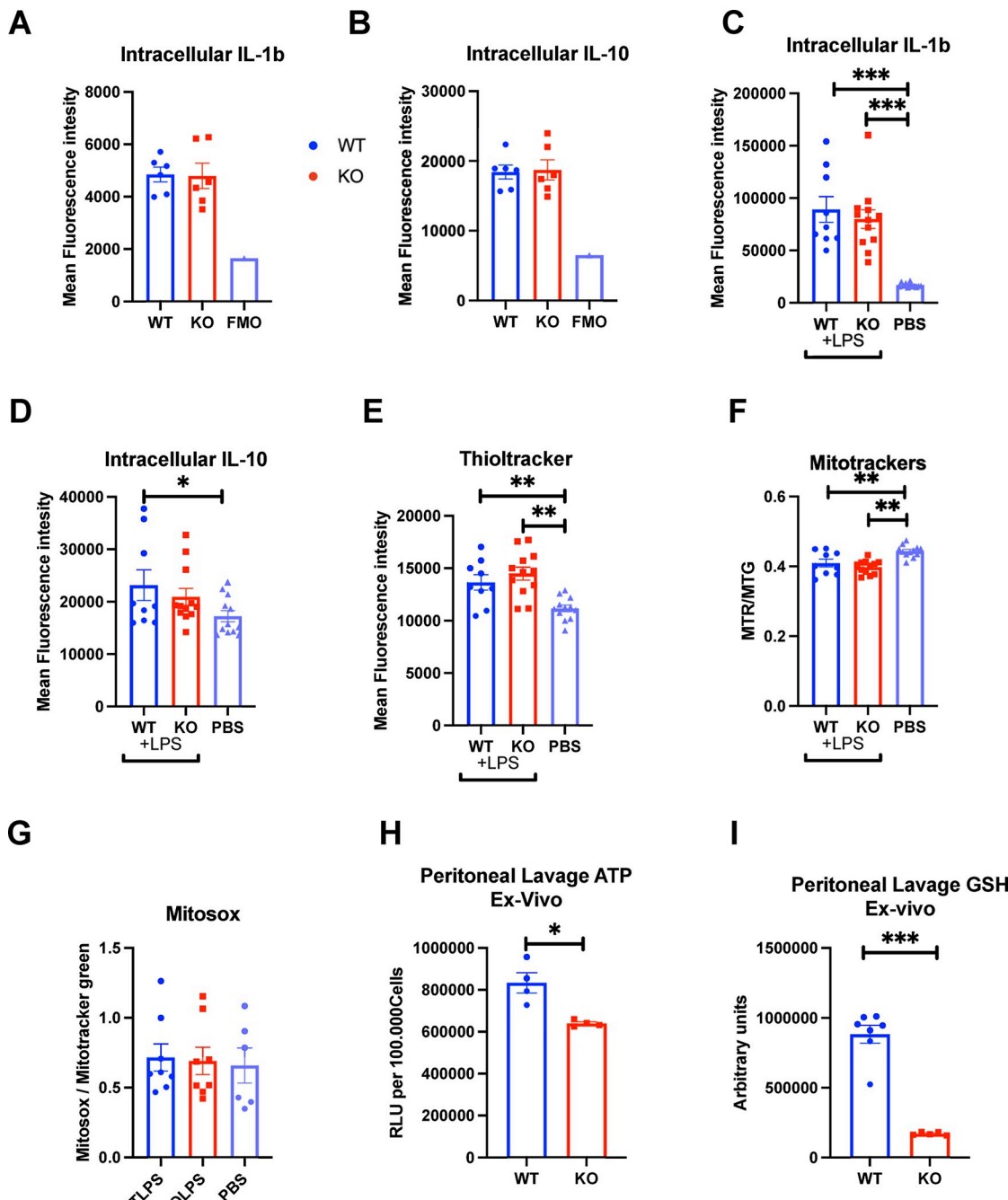

**Fig 5. *In vivo* assessment of peritoneal macrophage function.** (A) Intracellular IL-1b staining, (B) Intracellular IL-10 staining, (C) Intracelllular IL-1b and (D) IL-10 staining in macrophages from LPS pre-treated animals, (E) Thioltracker thiol-dependent fluorescence, (F) mitochondrial membrane potential, (G) mitochondrial superoxide production, (H) ATP and (I) GSH levels in peritoneal lavage, measured immediately after isolation. Each dot represents peritoneal lavage from 1 mouse. Statistics: two-sided Mann–Whitney U test; error bars represent SD; ★p < 0.05, ★★p<0.01, ★★★p < 0.001.

membrane potential group, KO macrophages exhibited lower membrane potential, while the inverse was observed in the low membrane potential population. This suggests that Sxc activity may have a modulatory role in mitochondrial activity, particularly since the exported glutamate can be derived from the Krebs cycle metabolite 2-oxoglutarate [12].

Functionally, KO BMDMs displayed reduced production and secretion of IL-1b and IL-10, as well as decreased intracellular cytokines, suggesting a possible deficiency of cysteine for protein synthesis. Treatment with the GSH precursor gamma-glutamyl-cysteine, which is readily taken up through an Sxc-independent mechanism, increased IL-1b production. Furthermore, Chac1, a gene encoding a glutathione-degrading enzyme that is upregulated upon cysteine shortage, is chronically upregulated in KO macrophages. Disruption of Chac1 in both WT and KO macrophages induced mild ER stress and paradoxically led to lower GSH levels, potentially due to disrupted GSH metabolism. This also increased IL-1b production in WT macrophages, possibly due to underlying dysfunction that elevates inflammatory tone. In KO macrophages, *Chac1* disruption decreased IL-1b production, suggesting that CHAC1 is not a limiting factor in cytokine production but rather a vital stress response to cystine starvation that improves function.

*In vitro*, peritoneal macrophages exhibited a similar phenotype as BMDMs. We then employed an endotoxemia model as a simulation of systemic bacterial infection with a ligand that heavily induces *Slc7a11* expression in macrophages. Sxc deficiency did not result in a difference in intracellular cytokines of isolated peritoneal macrophages with or without endotoxemia. This can be explained by the many differences between *in vitro* and *in vivo* settings, such as cross-talk with other cells and a completely different environment. We find the most convincing explanation the difference in atmospheric composition between cell culture and the peritoneal cavity [13]. Higher oxygen tension could, for example, shift the cystine/cysteine ratio towards its oxidized counterpart in the extracellular space [14]. This then results in increased Sxc substrate availability and thereby increased activity, leading to an overemphasis of importance compared to conditions in the peritoneal cavity. A similar exclusively *in vitro* relevance for Sxc has also been reported in T-cells [15]. Moreover, Sxc has been implicated in the wound healing response, demonstrating a strong effect *in vivo*. This is likely due to skin wounds being exposed to atmospheric oxygen tensions, suggesting that Sxc might be particularly important in the response to bacterial invasion of the skin surface [10]. Additionally the authors found that Sxc deficiency increased the phagocytic response, suggesting that Sxc activity cements an inflammatory non-efferocytotic phenotype in macrophages. This likely describes a trade-off between wound healing and pathogen vigilance that is modulated by Sxc.

This study illustrates the requirement of Sxc for the macrophage anabolic response to bacteria *in vitro*, with implications for GSH levels, cytokine production, and mitochondrial function. *In vivo*, this effect was not observed, likely due to the presence of numerous redundant metabolic pathways allowing for compensation. Different atmospheric conditions compared to *in vivo* settings might also skew the ratio of cysteine/cystine towards the latter, stimulating Sxc activity. It therefore seems probable that Sxc is especially important for the immune response in barrier tissues exposed to atmospheric oxygen tensions. Significant changes in cystine metabolism in response to LPS treatment suggest Sxc's involvement. In essence, this study highlights the importance of Sxc in a specific context while also illustrating the robustness of the metabolic system *in vivo* to perturbations of a single node in the network.

## Author Contributions

**Conceptualization:** Axel de Baat, Adriano Fontana.

**Data curation:** Axel de Baat.

**Formal analysis:** Axel de Baat.

**Funding acquisition:** Marc Y. Donath.

**Investigation:** Axel de Baat.

**Methodology:** Axel de Baat.

**Resources:** Axel de Baat.

**Software:** Axel de Baat.

**Supervision:** Daniel T. Meier, Marianne Böni-Schnetzler, Marc Y. Donath.

**Visualization:** Axel de Baat.

**Writing – original draft:** Axel de Baat, Daniel T. Meier, Adriano Fontana, Marianne Böni-Schnetzler.

**Writing – review & editing:** Axel de Baat, Daniel T. Meier, Adriano Fontana, Marianne Böni-Schnetzler, Marc Y. Donath.

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
