## [Decision Letter · Decision Letter 0]

11 Jul 2023

PONE-D-23-15179Cystine/glutamate antiporter system Xc- deficiency impairs macrophage glutathione metabolism and cytokine production.PLOS ONE

Dear Dr. de Baat,

Thank you for submitting your manuscript to PLOS ONE. After careful consideration, we feel that it has merit but does not fully meet PLOS ONE’s publication criteria as it currently stands. Therefore, we invite you to submit a revised version of the manuscript that addresses the points raised during the review process. Please submit your revised manuscript by Aug 25 2023 11:59PM. If you will need more time than this to complete your revisions, please reply to this message or contact the journal office at plosone@plos.org. Please include the following items when submitting your revised manuscript:A rebuttal letter that responds to each point raised by the academic editor and reviewer(s). You should upload this letter as a separate file labeled 'Response to Reviewers'.A marked-up copy of your manuscript that highlights changes made to the original version. You should upload this as a separate file labeled 'Revised Manuscript with Track Changes'.An unmarked version of your revised paper without tracked changes. You should upload this as a separate file labeled 'Manuscript'.

We look forward to receiving your revised manuscript.

Kind regards,

Subhadip Mukhopadhyay, PhD

Academic Editor

PLOS ONE

3. To comply with PLOS ONE submissions requirements, in your Methods section, please provide additional information regarding the experiments involving animals and ensure you have included details on (1) methods of sacrifice, (2) methods of anesthesia and/or analgesia, and (3) efforts to alleviate suffering.

5. Please upload a new copy of Figure 1 as the detail is not clear. Please follow the link for more information: https://blogs.plos.org/plos/2019/06/looking-good-tips-for-creating-your-plos-figures-graphics/" https://blogs.plos.org/plos/2019/06/looking-good-tips-for-creating-your-plos-figures-graphics/

Reviewers' comments:

Reviewer's Responses to Questions

**Comments to the Author**

1. Is the manuscript technically sound, and do the data support the conclusions?

Reviewer #1: Yes

Reviewer #2: Partly

Reviewer #3: Partly

2. Has the statistical analysis been performed appropriately and rigorously? 

Reviewer #1: Yes

Reviewer #2: Yes

Reviewer #3: Yes

3. Have the authors made all data underlying the findings in their manuscript fully available?

Reviewer #1: Yes

Reviewer #2: No

Reviewer #3: Yes

4. Is the manuscript presented in an intelligible fashion and written in standard English?

Reviewer #1: Yes

Reviewer #2: Yes

Reviewer #3: Yes

5. Review Comments to the Author

Reviewer #1: The manuscript sounds good and well written but sequences should be added to the manuscript

Results should be titled according to method's procedures

Product ID and any modifications or precautions done to procedures should be clarified

Reviewer #2: In this manuscript, the authors investigated the functional implication of glutamate-cystine antiporter (sXc) system in the regulation of macrophage responses. The manuscript is well-written with enough experimental details. Statistical analysis is done carefully and mentioned in the manuscript in a clear manner. However, it is concerning that the manuscript is missing figure 1 and there is duplication of figure 2 instead. Rest of the figures and the respective experimental analysis are quite convincing, and the authors tried to highlight the physiological implication/importance for each experiment. A list of corrections and suggestion regarding the manuscript are mentioned below:

1. In the manuscript, a space is required between effect and or in line 2 (introduction).

2. In line 11, authors mentioned that recent studies have shown how mitochondria modulates immune cell function by regulating metabolism and redox homeostasis but cited only one article. Why?

3. Similarly, citation related correction is needed in lines 23-26. Moreover, in this part of the introduction (and in the discussion), authors tried to draw a connection between their study and another publication that demonstrated the effect of sXc deficiency and inhibited sXc in improved wound healing in diabetic mouse. Additional clarification on this connection would be helpful for the general readers.

4. The end of the introduction is abrupt; hence, a brief (1-2 lines) description of the overall findings would make it complete.

5. A brief description of endotoxemia and how introduction of LPS would mimic the condition would help the readers to follow the study.

6. In line 82, the sentence ends abruptly with a question mark.

7. In line 84, the Cas9 concentration is written as pmol µM, which needs to be corrected.

8. The lack of figure 1 makes it difficult to assess the transcriptional regulation of sXc.

9. Legends are missing for figure 4a and 4b. Also, in figure 4b, the X-axis label WTLPS and KOLPS need to replaced with WT LPS and KO LPS.

10. Figure 5 is also missing the legend.

Overall this manuscript is incomplete without figure 1. The decision can only be made once the data representing figure 1 are submitted (only if it is permitted by the Editor) and thoroughly evaluated.

Reviewer #3: The main idea of the manuscript was investigated earlier and the involvement of the cystine transport system in macrophage activity was reported since 1994 by Piani and Fontana (an author of the current manuscript) and in other publications as well. However, I thoroughly enjoyed reviewing the manuscript and only have some minor requests for revision.

1- The details for both the ex vivo and in vivo experiments should be mentioned clearly in the materials and methods section

2- In line 82, "Product ID and sequences are listed in where?" should be removed

3- In line 92-93, "Journal probably wants article numbers/clones of Ab used here or supplemental table." should be removed

4- Both the ex vivo and in vivo results should be discussed and explained in more details in the discussion section

6. PLOS authors have the option to publish the peer review history of their article (what does this mean?). If published, this will include your full peer review and any attached files.

Reviewer #1: **Yes**

Reviewer #2: No

Reviewer #3: No

---

## [Author Response · Author response to Decision Letter 0]

12 Aug 2023

Reviewer #1:

We appreciate the positive feedback and have made the following revisions:

- We have now included sequences in the manuscript.

- Titles have been added to align more accurately with the method's procedures.

- Product ID and any modifications or precautions done to procedures have been clarified.

Reviewer #2:

Thank you for your valuable comments. We have responded as follows:

1. We corrected the spacing issue on line 2 of the Introduction.

2. We added more references to support the statement on line 11.

3. We have made the requested citation correction on lines 23-26 and provided additional clarification on the connection between our study and the publication demonstrating the effect of Sxc deficiency.

4. A brief description of the overall findings has been added at the end of the introduction.

5. A brief description of endotoxemia and the role of LPS has been included.

6. Line 82 has been rephrased for clarity.

7. The Cas9 concentration on line 84 has been corrected.

8. We have included Figure 1 in the revised manuscript.

9. Legends for Figure 4a and 4b have been added and labels on Figure 4b have been corrected.

10. Figure 5 now includes the legend.

Reviewer #3:

Thank you for your positive feedback and for your minor requests for revision:

1. We have provided detailed information on both the ex vivo and in vivo experiments in the Materials and Methods section.

2. The phrase, "Product ID and sequences are listed in where?" has been removed from line 82.

3. The phrase, "Journal probably wants article numbers/clones of Ab used here or supplemental table." has been removed from lines 92-93.

4. We have expanded our discussion and explanation of both the ex vivo and in vivo results in the Discussion section.

---

## [Decision Letter · Decision Letter 1]

10 Sep 2023

Cystine/glutamate antiporter system x_c_^-^ deficiency impairs macrophage glutathione metabolism and cytokine production.

PONE-D-23-15179R1

Dear Dr. Axel de Baat,

We’re pleased to inform you that your manuscript has been judged scientifically suitable for publication and will be formally accepted for publication once it meets all outstanding technical requirements.

Kind regards,

Subhadip Mukhopadhyay, PhD

Academic Editor

PLOS ONE

Additional Editor Comments (optional):

Reviewers' comments:

Reviewer's Responses to Questions

**Comments to the Author**

1. If the authors have adequately addressed your comments raised in a previous round of review and you feel that this manuscript is now acceptable for publication, you may indicate that here to bypass the “Comments to the Author” section, enter your conflict of interest statement in the “Confidential to Editor” section, and submit your "Accept" recommendation.

Reviewer #2: All comments have been addressed

Reviewer #3: (No Response)

2. Is the manuscript technically sound, and do the data support the conclusions?

Reviewer #2: Yes

Reviewer #3: (No Response)

3. Has the statistical analysis been performed appropriately and rigorously? 

Reviewer #2: Yes

Reviewer #3: (No Response)

4. Have the authors made all data underlying the findings in their manuscript fully available?

Reviewer #2: Yes

Reviewer #3: (No Response)

5. Is the manuscript presented in an intelligible fashion and written in standard English?

Reviewer #2: Yes

Reviewer #3: (No Response)

6. Review Comments to the Author

Reviewer #2: In this revised manuscript, the authors investigated the functional implication of the glutamate-cystine antiporter (sXc) system in the regulation of macrophage responses. The authors did a very careful job of editing and revising the content of the manuscript. The statistical analysis of the data was done thoroughly. Also, figure 1 was added properly, which supports their interpretation and the analysis of the transcriptional regulation of Sxc, Slc7a11. As requested, the authors included a very brief description of their overall findings at the end of the introduction, which might be useful for the general readers to get an overall impression of the study. The authors also added a short definition of the term endotoxemia. Finally, the authors made nearly all the corrections (textual, figure legends, figure labeling, etc.), which were needed for the publication. Overall, this manuscript reads well and would be a nice addition to the field of the cystine transport system in macrophage activity.

Reviewer #3: The authors has done a nice job of revising this manuscript and they have addressed all of my comments.

7. PLOS authors have the option to publish the peer review history of their article (what does this mean?). If published, this will include your full peer review and any attached files.

Reviewer #2: No

Reviewer #3: No

---

## [Editor Report · Acceptance letter]

25 Sep 2023

PONE-D-23-15179R1 

Cystine/glutamate antiporter system xc- deficiency impairs macrophage glutathione metabolism and cytokine production. 

Dear Dr. de Baat:

I'm pleased to inform you that your manuscript has been deemed suitable for publication in PLOS ONE. Congratulations! Your manuscript is now with our production department. 

Kind regards, 

on behalf of

Dr. Subhadip Mukhopadhyay 

Academic Editor

PLOS ONE